# Vaccine Hesitancy and Public Mistrust during Pandemic Decline: Findings from 2021 and 2023 Cross-Sectional Surveys in Northern Italy

**DOI:** 10.3390/vaccines12020176

**Published:** 2024-02-08

**Authors:** Verena Barbieri, Christian J. Wiedermann, Stefano Lombardo, Giuliano Piccoliori, Timon Gärtner, Adolf Engl

**Affiliations:** 1Institute of General Practice and Public Health, Claudiana—College of Health Professions, 39100 Bolzano, Italy; 2Department of Public Health, Medical Decision Making and Health Technology Assessment, University of Health Sciences, Medical Informatics and Technology, 6060 Hall, Austria; 3Provincial Institute for Statistics of the Autonomous Province of Bolzano—South Tyrol (ASTAT), 39100 Bolzano, Italy

**Keywords:** COVID-19 vaccination, compulsory vaccination, trust in institutions, language, CAM

## Abstract

This study examines vaccine agreements in South Tyrol, Italy, within distinct socio-cultural and linguistic contexts. Using data from the 2021 and 2023 “COVID-19 Snapshot Monitoring” extended surveys, we assessed changes in attitudes towards COVID-19 and other vaccinations during the second and final years of the pandemic. Multivariate logistic regression analysis was used to examine factors such as trust in institutions, language groups, and the use of complementary and alternative medicine. The representativeness of the study is supported by good participation rates, ensuring a comprehensive view of attitudes towards vaccination in the region. The results show a shift in public agreement with the national vaccination plan to 64% by 2023, from a rate of about 73% agreement in 2021 (*p* < 0.001). A significant decrease in trust in health authorities and a negative correlation with complementary and alternative medicine consultations were observed. The results highlight the complex nature of vaccine hesitancy in diverse regions such as South Tyrol and underline the need for targeted communication strategies and trust-building initiatives to effectively reduce hesitancy. This study provides critical insights for the formulation of public health strategies in diverse sociocultural settings.

## 1. Introduction

The severe acute respiratory syndrome coronavirus 2 (SARS-CoV-2) pandemic, manifested as coronavirus disease 2019 (COVID-19), has highlighted the critical role of vaccination in public health [1,2]. Although safe and effective COVID-19 vaccines have been rapidly developed, vaccine hesitancy and refusal have emerged as significant global challenges, affecting the success of vaccination campaigns and public health efforts [3,4,5]. Vaccine hesitancy, defined as a delay in accepting or refusing vaccines despite the availability of services, is influenced by factors such as mistrust in vaccines or health systems, doubts about safety, and misinformation [5,6]. This complexity is further exacerbated by the different vaccination policies adopted by different countries, ranging from mandatory policies in some to voluntary approaches in others [6,7]. Understanding the reasons and characteristics of vaccine acceptance is essential for designing effective communication strategies and interventions to increase vaccine uptake.

During the COVID-19 pandemic, vaccine hesitancy emerged as a significant challenge that has directly affected public health strategies and efforts to achieve herd immunity. The profound impact of vaccine hesitancy on individual and community health underscores the need to understand and address the underlying causes to ensure the effectiveness of vaccination programs [8]. The “COVID-19 Snapshot Monitoring” (COSMO) survey [9], a European instrument that has been repeated in several countries including Italy [10] since 2021, has played a key role in assessing public attitudes towards pandemic-related health measures. In response to the historically high level of vaccine hesitancy in South Tyrol, Italy [11,12], the 2021 application of the COSMO survey in this northern Italian province included additional questions to improve our understanding of public trust in information sources, perceptions of health institutions, the role of altruism, and attitudes towards vaccines [13,14,15].

The main objectives of the first and second surveys carried out in 2021 [13,14,15] and 2023 in South Tyrol were to explore the dynamics of vaccine acceptance in a region characterized by low vaccine uptake [16] against the background of linguistic and cultural diversity [17]. The 2023 survey added new elements to better understand the acceptance of vaccines. These include complementary and alternative medicine (CAM) consultations [18], as this additional factor is relevant to understanding the complex nature of vaccine hesitancy [19]. In 2021, the COVID-19 vaccine hesitancy rate in South Tyrol was approximately 15% [13], which is in line with the hesitancy rates reported in other regions [20], where traditional non-COVID-19 vaccination rates were generally much higher than in South Tyrol [11,12]. Thus, the comparable hesitancy rate is in marked contrast to the lower vaccination rates in South Tyrol, both during and before the pandemic.

The unique circumstances and uncertainties created by COVID-19 may positively influence attitudes towards vaccination [21]. Therefore, a new survey was conducted in 2023 during the post-pandemic period to further explore this possible shift in South Tyrol. The 2023 survey aimed to assess whether the trends observed in 2021 persisted or evolved after the peak of the pandemic, and to understand the factors contributing to any changes in vaccine acceptance and uptake. The results of this survey are expected to provide valuable insights into the evolving dynamics of vaccine acceptance and contribute to a broader understanding of how public health crises can affect vaccine perceptions and behavior.

## 2. Materials and Methods

### 2.1. Study Design and Data Collection

This investigation utilized a cross-sectional survey design based on random sampling. The Autonomous Province of Bolzano–South Tyrol Statistical Institute (ASTAT) engaged in the recruitment of a stratified random sample from the South Tyrol populace, explicitly excluding individuals residing in nursing homes. This stratification was executed across various demographics, including municipality, gender, and age categories (18–34, 35–49, 50–64, and 65+ years), employing the “Surveyselect” function within SAS v9.2 for sample selection. All participants were required to be 18 years of age or older. The inaugural survey took place in March 2021, as outlined in previous documentation [13], and was subsequently replicated in February 2023. Data handling by ASTAT was carried out in strict adherence to the EU General Data Protection Regulation.

For both surveys, approximately 4000 out of the 430,000 adult residents of South Tyrol were selected to participate through a one-stage stratified random sampling method tailored for this quantitative analysis. In 2021, 4400 postal invitation letters were sent, eliciting 1425 responses (32.4%). In 2023, the number of invitation letters sent was 3800, from which we received 1388 responses (36.5%). Samples were gathered independently to maintain participant anonymity. The determination of the sample size was informed by an anticipated participation rate of 33%, which is consistent with rates observed in earlier surveys within this series of studies [22].

Invitations to participants were dispatched through letters, which specified the intended date of participation, provided a link to the online survey (accompanied by telephone assistance) that explored demographic, clinical, and socio-behavioral dimensions, and included a unique password to serve as a code for pseudo-anonymization.

### 2.2. Questionnaire

The questionnaire was an extended version of the COSMO Italy and COSMO Germany surveys [10,23]. Its repeated administration was part of an Italian survey in 2021, whereas, in 2023, the survey was conducted only in South Tyrol. Questions on vaccine hesitancy were added to the official WHO questionnaire used in COSMO [24], including items measuring trust in local information sources and institutions [25,26], conspiracy perceptions [27], altruism [28], and CAM consultation [29,30] in 2023.

Sociodemographic inquiries were tailored to fit the unique context of South Tyrol, encompassing questions specific to the municipality and native language (German, Italian, Ladin, among others).

### 2.3. Agreement with the National Vaccination Plan (Dependent Variable) and Further Questions about Vaccine Acceptance

Agreement with the national vaccination plan was measured using the question “Do you agree with the national vaccination plan?” on a 6-point Likert scale (1 = strongly disagree to 6 = strongly agree), while COVID-19 vaccine hesitancy was measured using the dichotomous question, “Would you be vaccinated against COVID-19?” in 2021 and “How many times have you been vaccinated against COVID-19?” in 2023.

The question “Has the pandemic changed your attitude towards vaccination?” was also asked with 3 possible answers: “No”, “Yes, I support it more now”, and “Yes, I support it less now”. To obtain more detailed information, questions on trust in vaccination, beliefs about the COVID-19 vaccine itself, and opinions about the COVID-19 vaccination were added.

### 2.4. Putative Predictors of Agreement with the National Vaccination Plan (Independent Variables)

Sociodemographic variables were utilized to forecast agreement with vaccination. Factors such as age, gender, native language (German, Italian, Ladin, or other/multiple languages), residential area, education level on a four-point scale, possession of Italian citizenship, employment in the healthcare sector (as binary data), presence of chronic illnesses, and economic status over the past three months (evaluated on a three-point scale with an additional “don’t know” option) were included. Furthermore, variables influencing COVID-19 vaccination consent were identified from the existing literature and the COSMO survey, including confidence in information sources and institutions [25,26] (such as health authorities and government bodies), which was assessed using a 6-point Likert scale from 1 = “no trust at all” to 6 = “high trust”, plus an option for “don’t know”. In addition, we measured conspiracy beliefs (five questions on a 6-point Likert scale from 1 = “strongly disagree” to 6 = “strongly agree”) [27], altruism (five questions on a 6-point Likert scale from 1 = “don’t agree at all” to 6 = “completely agree”), and CAM consultation within the past 12 months [29,30]. All variables were considered predictors of agreement with the national vaccination plan.

### 2.5. Statistical Analysis

Quantitative data are presented by the median, along with the first and third quartiles. Given the non-normal distribution of all quantitative variables, the Mann–Whitney U test was used to determine significant differences between groups. Categorical and ordinal variables are presented as counts and proportions. Group comparisons were performed using the chi-square test, and the Kendall tau-b test was used to assess correlations. In both 2021 and 2023, cumulative scores were determined for belief in conspiracy theories (with Cronbach’s alpha values of 0.81 and 0.83, respectively), levels of altruism (with Cronbach’s alpha values of 0.79 and 0.77, respectively), and trust in institutions (with Cronbach’s alpha values of 0.92 and 0.93, respectively).

Regarding trust in institutions, the “I don’t know” response was initially assigned a value of 3.5 (the midpoint of the 1–6 scale) and then considered “missing” in a separate analysis. With this latter method, a participant’s total score was calculated only if responses to all relevant questions were available, with a higher total score reflecting increased trust. Logistic regression analysis was used to model compliance with the national vaccination strategy using several predictor variables. The analysis distinguished between variables that were statistically significant and those that were not. The linearity of continuous variables was assessed by including a quadratic term. Model diagnostics included the use of the DFBETA statistic, Cook’s distance, and the identification of leverage points to assess the influence on the model. The overall quality and predictive accuracy of the model were assessed by ROC curve analysis of predicted values. Bujang and colleagues [31] suggested a minimum of 500 participants as the required sample size for observational studies in large populations. In addition, they advocate the use of the formula n = 100 + i × 50, where “i” represents the number of independent variables. Thus, for a model with 11 independent variables, the essential sample size is calculated as n = 100 + 11 × 50 = 650. Traditional sample size calculations, assuming a 30% event probability and an odds ratio (OR) of 1.2, along with a 5% type I error rate and 90% power, required a sample size of n = 1239. This calculation was facilitated by GPower version 3.1.9.4. *p*-values less than 0.001 are indicated by ***, less than 0.01 by **, less than 0.05 by *, and *p*-values of 0.05 or greater are indicated as not significant (n.s.). SPSS version 27 was used for all statistical analyses.

## 3. Results

### 3.1. Sample Characteristics

The response rate was 32% in 2021 and 39% in 2023. The demographic characteristics of the dataset were representative of age, sex, municipality, and native language. Table 1 lists the sample characteristics for the two years. Only the economic situation in the last three months (*p* = 0.048) and level of education (*p* = 0.041) were slightly significantly different between the two surveys.

### 3.2. Agreement with the National Vaccination Plan and COVID-19 Vaccine Uptake

In the March 2021 survey, 73% of the participants agreed with the national vaccination plan, whereas in 2023, only 64% agreed (*p* < 0.001). Agreement was significantly positively correlated with increasing age in both years (Kendall tau-b 0.199 and 0.158; *p* < 0.001, respectively) (Figure 1). The overall percentage of people with COVID-19 vaccine uptake by 6 September 2021 [32] was approximately 68% and increased significantly with increasing age (*p* < 0.001), whereas the percentage of people with COVID-19 vaccine uptake of at least one dose (92.1%; Kendall tau-b 0.063; *p* 0.002) and at least two doses (89.4%; Kendall tau-b 0.096; *p* < 0.001) in 2023 increased only slightly with age.

### 3.3. Perceptions of Vaccination in the Light of the Pandemic

#### 3.3.1. General Vaccination

Respondents were asked if their views on mandatory vaccination against viruses other than coronaviruses had changed as a result of the pandemic. A significant majority, 73.8% in 2021 and 75.3% in 2023, reported no change in their position on mandatory vaccination for other viruses in both years, with the difference between the two years not statistically significant; 19.6% said they were more supportive in 2021 because of the pandemic compared with 15.1% in 2023 (*p* < 0.001). The corresponding percentages of those who now support it are 6.6% in 2021 and 9.6% in 2023. This change was significantly different (*p* < 0.001) between those who agreed to the national vaccination plan and those who did not (Figure 2).

In 2021, 88% of parents with children aged 0–6 years said they would vaccinate their child, and by 2023, 85% of parents with children aged 0–6 years and 88% of all concerned people said they would agree to childhood vaccination. The differences between agreeing and disagreeing with the national immunization plan are highlighted in Figure 2. The differences between agreeing and disagreeing participants were highly significant (*p* < 0.001) in both years. No significant differences were found between the years.

#### 3.3.2. COVID-19 Vaccinations

In 2021, 16% of the participants said they would not be vaccinated against COVID-19, and 84% said they would. In 2023, only 8% said they had never been vaccinated against COVID-19 and 3% said they had only been vaccinated once. In addition, 19% reported having been vaccinated twice, 57% three times, and 13% recommended four times.

The differences between the participants who agreed with the national vaccination plan and those who disagreed are shown in Figure 2. The differences between the agreeing and disagreeing participants were highly significant (*p* < 0.001 for both years).

#### 3.3.3. Influenza Vaccination

In 2023, 18.7% of the participants said they had been vaccinated or were thinking of being vaccinated against influenza, 73.8% said they had not been vaccinated, and 7.8% did not know. The responses differed significantly (*p* < 0.001) between those who supported the national vaccination plan (yes: 24.1%; no: 65.7%; do not know: 10.2%) and those who did not (yes: 8.7%; no: 87.9%; do not know: 3.4%).

#### 3.3.4. Reasons for COVID-19 and Compulsory Vaccinations

In 2021, participants agreed significantly more with decisions regarding COVID-19 vaccination and other vaccinations made by the authorities than in 2023 (Table 2). Trust in COVID-19 vaccination was significantly lower in 2023 for all questions. The necessity of COVID-19 vaccination was seen to be significantly less in 2023 for all questions, despite the question “COVID-19 vaccination is not necessary because natural herd immunity is achieved with virus spread and this is quite sufficient”. However, there was no difference between the questions regarding the harmfulness of the COVID-19 vaccination between the two years.

There are no significant differences between 2021 and 2023 regarding statements about the necessity of compulsory childhood vaccination. Statements regarding harmfulness did not differ significantly, with the exception of the statement, “... there have been negative vaccine experiences in my family” (2021: 11.2% vs. 2023: 23.1%, *p* = 0.011). Compulsory vaccination in light of COVID-19 was considered significantly less important in 2023 than in 2021 for all questions, while no significant difference was found for the questions regarding the reasons for vaccinating children.

#### 3.3.5. Perceived Problems Due to the Lack of a “Green Pass”

The question “Did you have problems because you did not immediately get or did not get the green pass?” can be responded to using multiple answers. Most participants stated that they had problems at work (12%), followed by problems during their free time (11%). Only 2% had problems with their family or friends. A total of 82% of participants stated that they had no problems. People who did not agree to the national vaccination plan had significantly more problems in any field (*p* < 0.001, each). The details are shown in Figure 3. People who did not receive COVID-19 vaccination at all or who were vaccinated only once had significantly more problems in all five fields (*p* < 0.001, each).

### 3.4. Multivariate Logistic Regression to Predict Agreement with the National Vaccination Plan

A logistic regression model was used to identify independent predictors of compliance with the national vaccination plan (Table 3). Significant demographic predictors were age as a continuous factor (2021: Kendall tau-b = 0.069 **; 2023: 0.073 **), urban residence (2021: Kendall tau-b = 0.054 *; 2023: 0.092 **), native language (2021: Kendall tau-b = 0.078 **; 2023: 0.103 **) as a categorical variable, and economic situation in the last three months (2021: Kendall tau-b: −0.097 ***; 2023: −0.100 ***). The educational level was significant in 2021 only. Other predictors were the dichotomous variable “no COVID-19 vaccination” (2021: Kendall tau-b = −0.494 ***; 2023: −0.308 ***), total scores for trust in institutions (2021: Kendall tau-b = 0.297 ***; 2023: 0.364 ***), altruism (2021: Kendall tau-b = 0.073 **; 2023: 0.062 **) and, in 2023, a dichotomous variable indicating whether participants had consulted a CAM provider in the past 12 months (Kendall tau-b: −0.139 ***).

For each model, the linearity of continuous predictors was assessed by incorporating their quadratic terms, which were found not to be statistically significant. The models utilized data from N = 1425 participants in 2021 and N = 1388 in 2023. Further analysis, adhering to the methodology section and considering only complete sum scores (N = 920 in 2021 and N = 982 in 2023), yielded consistent outcomes for 2023, with the detailed findings not presented here. However, this approach did not align well with the 2021 data. In 2021 and 2023, trust in institutions was confirmed as a significant positive predictor of agreement with the national vaccination plan as well as COVID-19 vaccination and, in 2023, the Italian native language (using the German native language as an indicator). Furthermore, by 2023, consultation with a CAM provider was a negative predictor of agreement with the national vaccination plan.

The logistic regression model had an overall Nagelkerkes R^2^ of 0.341 in 2021 and 0.318 in 2023, and an overall model quality of 79.2% in 2021 was estimated using ROC analysis (area under the curve 0.792 [0.763; 0.821]) and 78.3% in 2023 (area under the curve 0.783 [0.760; 0.807]). Model diagnostics using DFBETA statistics did not reveal any outliers; Cook distance and leverage points showed that the models were stable and did not change after excluding single cases.

## 4. Discussion

Drawing on the expanded COSMO surveys conducted in 2021 and 2023 in South Tyrol, this study explored vaccine hesitancy within a region marked by its unique cultural and linguistic identity. The results show a shift in public agreement with the national vaccination plan from 73% by 2021, to a rate of about 73% agreement in 2023. General vaccination disagreement significantly increased by 2023, reflecting the complex interplay of public perception, trust, and pandemic-influenced attitudes. This research highlights not only the demographic determinants but also the significant role of diminishing trust in public health authorities and persistent concerns about vaccine safety. Moreover, it captures shifts in parental attitudes towards compulsory childhood vaccinations, underscoring the pandemic’s far-reaching impact on public health. These insights are essential for understanding the multifaceted nature of vaccine hesitancy and guiding future health communication and policy strategies.

As the COVID-19 pandemic wanes in 2023, it is important to distinguish between vaccine hesitancy due to mistrust and hesitancy due to diminished pandemic concerns. At the peak of the pandemic, the immediate threat of COVID-19 increased vaccine awareness. However, as this threat recedes, public perception of the need for vaccination may diminish, not necessarily because of mistrust, but because of a perceived reduction in risk. This change in public risk perception suggests that public health strategies need to adapt. Ongoing education and communication should focus on the long-term benefits of vaccination beyond the immediate pandemic context. This shift highlights the importance of regularly monitoring public attitudes toward vaccination to ensure that public health policies remain responsive and effective.

The observed decline in public agreement with decisions made by authorities regarding COVID-19 vaccination from 2021 to 2023 is a concerning trend that underscores a growing skepticism or change in public perception [33,34]. Initially, a substantial majority showed confidence in public health decisions, but the noticeable decrease over two years suggests an erosion of trust, possibly fueled by ongoing debates, evolving pandemic dynamics, and the public discourse surrounding vaccine efficacy and policy decisions.

This decline in trust extends to compulsory vaccinations beyond COVID-19, reflecting a broader apprehension towards governmental health mandates. The global decline in childhood vaccination coverage [35], particularly in Europe [36,37,38], post-pandemic, sets a concerning backdrop for understanding the specific vaccination challenges in South Tyrol. South Tyrol’s complex history, marked by its linguistic diversity and historical conflicts [17], significantly influences current vaccination attitudes and behaviors in the region [13].

Despite the global effort to underscore the efficacy and importance of COVID-19 vaccines, the increase in the number of individuals doubting the vaccine’s effectiveness from 2021 to 2023 indicates a deep-rooted skepticism. Concerns about the long-term risks and the novel nature of RNA vaccines have remained notably stable, suggesting persistent fears and apprehensions regarding vaccine safety. These findings highlight the critical role of addressing misinformation and providing clear and accessible information regarding vaccine safety and development processes [39]. The decrease in extreme denialist views about the existence of COVID-19 and its severity is a positive shift, yet the continued belief in conspiracy theories [40] points to the complex landscape of vaccine hesitancy, where scientific reassurance must be coupled with addressing underlying cultural and societal narratives [41].

The significant decrease in parents’ belief in the importance of protecting children through vaccination and ensuring herd immunity is alarming [42]. This shift might reflect the heightened anxieties and uncertainties brought about by the pandemic, potentially affecting long-standing attitudes towards routine childhood vaccinations. The increased reporting of negative vaccine experiences in families further complicates this scenario, potentially leading to a more personal and anecdotal basis for vaccine hesitancy [19]. The data indicate a complex evolution in public attitudes towards vaccination during and after the COVID-19 pandemic [42]. While the majority consistently expressed no change in their stance towards mandatory vaccinations for diseases other than COVID-19, the subtle shifts observed suggest an underlying reassessment of health priorities and trust in public health guidance over time in South Tyrol. The initial surge in support for vaccinations at the peak of the pandemic appears to wane slightly as the immediate threat recedes, reflecting a possible return to pre-pandemic perceptions or a new equilibrium in public health beliefs. This nuanced change underscores the importance of understanding long-term public sentiment trends, particularly because they may signal shifting priorities or emerging concerns that could influence future vaccination campaigns.

The significant correlation between individuals’ attitudes towards vaccinations and their stance on the national vaccination plan reveals the profound impact of health policies and public messaging [43]. This correlation is evident not only for COVID-19 vaccinations but also for general and influenza vaccinations, suggesting that people’s trust in and perception of public health policies can strongly influence their health-related decisions. As the pandemic continues to shape public health landscapes, this relationship underscores the need for clear, consistent, and transparent communication between health authorities. Tailored messaging that acknowledges and addresses specific community concerns, cultural contexts, and misinformation can be crucial in maintaining and boosting public trust and vaccine uptake.

The “Green Pass” items in the current survey showed that people who had not been vaccinated or had only been vaccinated once against COVID-19 had the most problems at work and in their leisure time and that people who had not been vaccinated at all had problems even within the family. This suggests that the discussion about green cards led to a higher level of mistrust [44,45]. In Italy, the most common concerns about the COVID-19 vaccine were its safety and efficacy and the “Green Pass” requirement [46]. A study among older people showed that the implementation of the “Green Pass” needs to be accompanied by effective information strategies [47], and it has been concluded elsewhere that to reduce misperceptions about the social norm of vaccination, governments and the media should report not only on the current COVID-19 vaccination rate but also on vaccination intentions and approvals in the early stages of the vaccination campaign [48].

The impact of COVID-19 on vaccine acceptance is another layer of complexity. Although the pandemic has heightened awareness of the importance of vaccination globally, its influence varies across regions. In South Tyrol, the initial heightened sense of urgency may have temporarily reduced hesitancy rates. However, as the immediate threat subsided, a slight rebound in hesitancy became evident. This pattern contrasts with regions in which sustained high levels of trust in public health measures may have led to a more stable decrease in hesitancy. The pandemic has undoubtedly acted as a catalyst, exposing and amplifying existing mistrust and skepticism towards health authorities. It is imperative for future research and policymaking to consider these regional disparities and the lessons learned during the pandemic to build more resilient and responsive public health systems that can effectively navigate the challenges of vaccine acceptance in the post-pandemic world.

In contrast to South Tyrol’s lack of vaccine acceptance in similar regions, particularly Trentino, the neighboring province in northern Italy, a clear picture emerges of how unique cultural and linguistic landscapes shape public health responses differently in the two neighboring provinces in Italy’s north. Both regions share certain geographical and socioeconomic characteristics, yet they exhibit distinct vaccine uptake patterns [11]. Comparative data illustrate that South Tyrol’s vaccination rates, apparently influenced by initiatives like the “Green Pass” and mandatory vaccinations [48,49,50,51,52], significantly diverge from those in neighboring Trentino and the national average, reflecting its unique public health landscape. In South Tyrol, the German-speaking [13], predominantly rural [15], population tends to exhibit lower rates of acceptance than Italian speakers.

Additionally, reliance on different information sources between linguistic groups in South Tyrol further contributes to divergence in vaccine hesitancy rates. The tendency of German-speaking South Tyroleans to access German or Austrian health information online [53] might lead to discrepancies in understanding and adhering to Italian vaccination policies, potentially fueling vaccine hesitancy. These observations underscore the critical need for health communication strategies that are not only linguistically tailored but also culturally sensitive to effectively address the concerns of diverse communities.

The uptake of or willingness to vaccinate against influenza was approximately 20% in our survey. This rate corresponds to the influenza vaccination rate in Italy [54] and Trentino [11] but not to the published data on vaccination in South Tyrol for the 2022/23 season, which is approximately 11% for the whole population [11]. Therefore, it can be assumed that half of the South Tyrolean population, who had a good willingness to be vaccinated against influenza, did not consider it important enough to do so during the pandemic. According to Porreca and Di Nicola [55], the COVID-19 era has led to higher coverage of influenza vaccination in Italy. However, in contrast to the national Italian general population, the regional level of influenza vaccination was not affected by the pandemic in South Tyrol, and the historical differences between Italian regions remained essentially unchanged. Despite the increase in COVID-19 vaccinations, the relatively low rate of influenza vaccination presents an ongoing public health challenge. This disparity highlights the necessity of distinguishing between vaccines and understanding the unique factors that drive vaccination against various diseases.

The logistic regression analysis from the study indicated the critical finding of a significant role of trust in institutions, which consistently emerged as an increasing predictor of agreement with the national vaccination plan for both 2021 and 2023. This underscores the importance of public trust in health authorities and policies [56,57,58]. Building and sustaining this trust is crucial for successful vaccination campaigns as it directly influences individuals’ willingness to comply with public health recommendations. The significant role of the native language, particularly Italian over German, suggests that health communication and policymaking must be culturally sensitive and linguistically tailored to effectively reach and resonate with diverse communities.

In 2023, the study also identified consulting with a CAM provider as a negative predictor of agreement with the national vaccination plan [18]. This reflects a broader challenge in public health: addressing diverse health beliefs and practices that may diverge from mainstream medical advice. The non-significant predictors, including age, chronic disease, working in the health sector, altruism, and economic situation, suggest that while these factors may influence general health behaviors, they do not have a direct impact on attitudes towards national vaccination policies in this context. Identifying consultation with CAM providers as a negative predictor of vaccination agreements underscores the necessity of engaging and addressing the concerns of those who seek alternative medical advice. The use of CAM presents both a challenge and an opportunity for public health policymakers and healthcare providers. The negative association between CAM consultation and national vaccination plans suggests that individuals who prefer CAM may have divergent health beliefs that influence their views on vaccination. 

The study on vaccine acceptance in South Tyrol has several limitations. First, the repeated cross-sectional design of the surveys limited their ability to establish causality between the identified factors and vaccine acceptance. Longitudinal studies with the same participants are required to observe changes over time more reliably and better understand the causative relationships. Second, the study relied on self-reported data, which might be subject to biases, such as social desirability or recall bias, potentially influencing the accuracy of the responses regarding vaccination attitudes and behaviors. Focusing on South Tyrol while providing in-depth regional insights might limit the generalizability of the findings to other regions with different cultural, linguistic, or healthcare contexts. Furthermore, while this study captures a snapshot of attitudes during the COVID-19 pandemic, these attitudes are likely dynamic and may evolve further as the situation changes, suggesting the need for ongoing research to capture these shifts. This study’s reliance on Italian- and German-speaking populations may not fully represent the diversity within South Tyrol, particularly the smaller Ladin-speaking community and other minority groups, potentially overlooking nuanced differences within these groups. Lastly, the quantitative nature of the study provides valuable statistical associations but may not capture the depth of individual motivations and concerns that qualitative approaches can offer. Future studies might benefit from a mixed-methods approach that combines quantitative surveys with in-depth interviews or focus groups to provide a more holistic understanding of vaccine hesitancy and acceptance in the region.

## 5. Conclusions

This study provides a comprehensive analysis of vaccine acceptance in South Tyrol, a region characterized by its unique cultural and linguistic diversity. The findings reveal a differentiated landscape in which vaccine hesitancy is not only influenced by individual beliefs and demographic factors but is also significantly shaped by cultural identity, language, and trust in public health authorities. There has been a marked negative shift in attitudes towards vaccination during this period, particularly in response to the COVID-19 pandemic and related public health measures. The findings of this study underscore the importance of culturally and linguistically tailored communication strategies to address and mitigate vaccine hesitancy effectively. Building trust in healthcare institutions and ensuring accessible and transparent information are critical for fostering public confidence and compliance with vaccination programs. In addition, understanding and addressing the concerns of those who consult CAM providers are critical for comprehensive public health strategies.

As South Tyrol addresses its unique public health challenges, the findings of this study can inform future policies and initiatives aimed at improving vaccination rates and public health outcomes. Continued research and monitoring are essential to adapt strategies to changing circumstances and ensure that all communities in South Tyrol are effectively reached and supported by their health needs. Lessons learned from South Tyrol’s experiences can also provide valuable insights into other regions with similar cultural and linguistic complexities.

## Figures and Tables

**Figure 1 vaccines-12-00176-f001:**
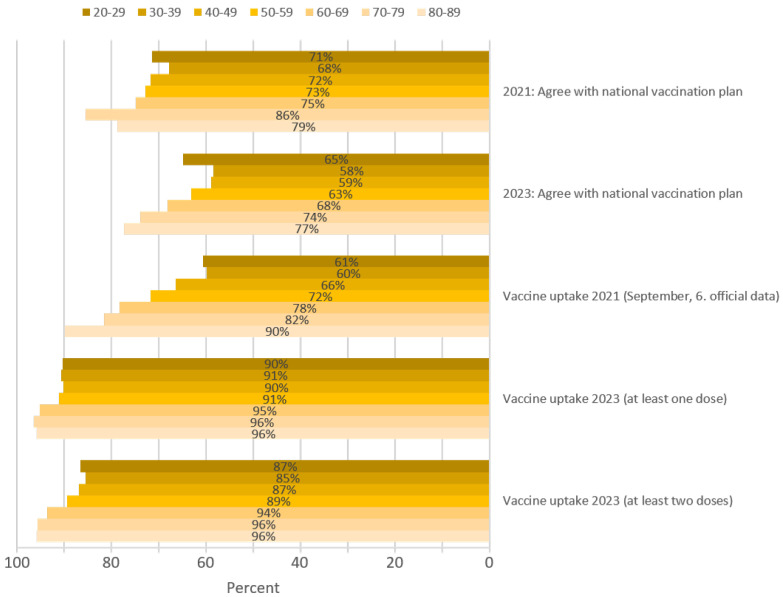
Agreement with the national vaccination plan in 2021 and 2023 and COVID-19 vaccine uptake in September 2021 [32] and in March 2023 per age group in South Tyrol.

**Figure 2 vaccines-12-00176-f002:**
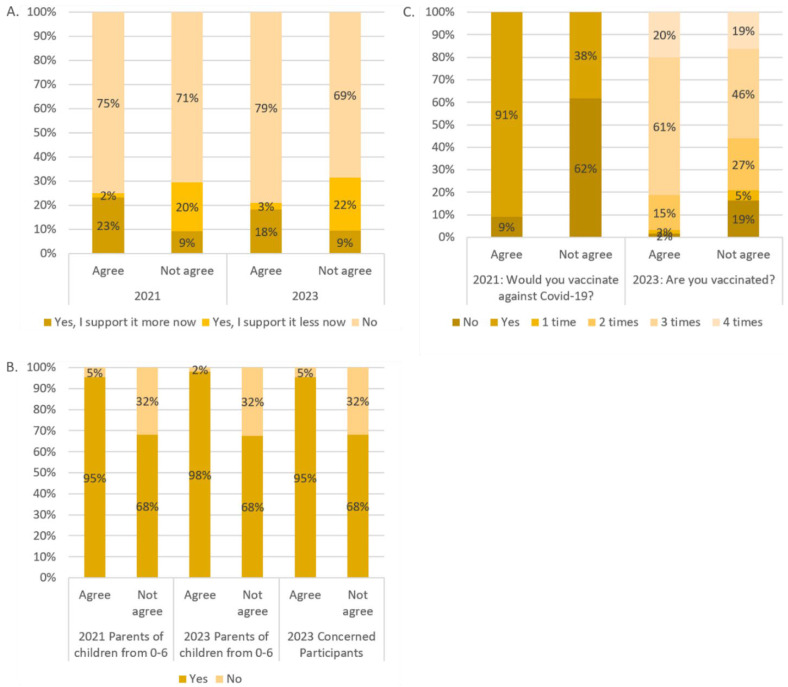
Agreement to the national vaccination plan for participant responses to the questions “Has the pandemic changed your attitude towards compulsory vaccination?” (**A**); “Would you vaccinate your child?” (**B**); and two questions about COVID-19 vaccination (**C**).

**Figure 3 vaccines-12-00176-f003:**
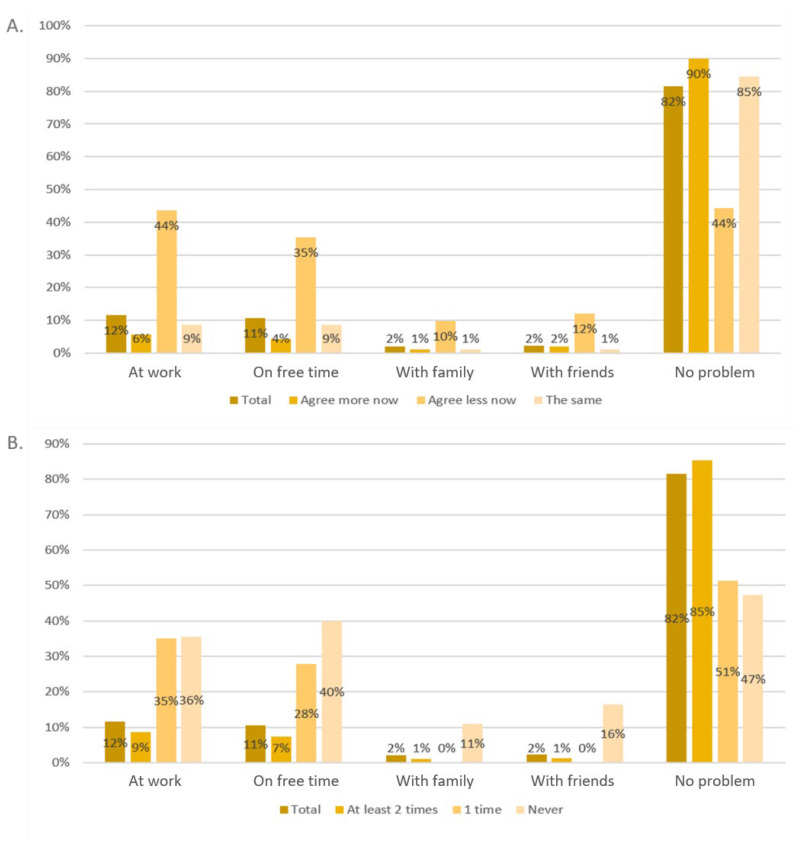
Problems because of a lack of a "Green Pass" for the questions “Do you agree with the national vaccination plan?” (**A**) and “How many times did you vaccinate against COVID-19?” (**B**).

**Table 1 vaccines-12-00176-t001:** Characteristics of the sample and comparison between non-hesitant and hesitant individuals.

Characteristics	2021 n = 1425 (100%)	2023 n = 1388 (100%)	*p*-Values ^†^
Age			
Years (median [1Q; 3Q])	50.0 [35; 64]	50.7 [36; 63]	n.s.
Gender			
Female (%)	51.5	51.0	n.s.
Education			
Middle school or lower (%)	22.2	18.1	0.041
Vocational school (%)	28.8	28.7	
High school (%)	28.8	31.3	
University (%)	20.1	21.8	
Residence			
Urban (%)	42.2	40.5	n.s.
Citizenship			
Italian (%)	91.7	90.2	n.s.
Native language ^‡^			
German (%)	61.7	63.1	n.s.
Italian (%)	26.9	27.1	
Ladin (%)	4	3.7	
Other/more than one (%)	7.4	6.1	
Working in the health sector			
Yes (%)	6.0	7.3	n.s.
Chronic disease(s)			
Yes (%)	17.3	18.2	n.s.
Economic situation (last 3 months)			
Better (%)	3.0	5.0	0.048
The same (%)	68.2	66.9	
Worse (%)	26.3	25.2	
Do not know (%)	2.5	3.0	

^†^ The *p*-values refer to chi-square tests for ordinal and nominal data and the Mann–Whitney U test for metric data. ^‡^ Native languages of South Tyrolean inhabitants. Abbreviation: n.s., not significant.

**Table 2 vaccines-12-00176-t002:** Attitudes towards COVID-19 vaccine and vaccination.

Category	Question	Response: Rather Agree or Agree	2021 N (%)	2023 N (%)	*p*-Value ^1^
			Total 1425 (100)	Total 1388 (100)	
	I think that decisions about vaccination against COVID-19 made by the public authorities are right	Yes	992 (70)	828 (60)	<0.001
		No	331 (23)	476 (34)	
	I think that decisions about mandatory vaccination (not COVID-19) made by the public authorities are right	Yes	952 (67)	837 (60)	<0.001
		No	348 (24)	440 (32)	
Trust in COVID-19 vaccination	I believe the vaccination can contain the spread of the virus ^1^		1227 (86)	979 (71)	<0.001
	If all others are vaccinated against the virus, I don’t need to get vaccinated		220 (15)	142 (10)	<0.001
COVID-19 vaccination is not necessary because…	…it is not effective		197 (14)	295 (21)	<0.001
	…natural herd immunity is achieved with virus spread and that is quite sufficient		274 (19)	288 (21)	n.s.
	…this disease does not exist/is a normal flu		150 (10)	65 (5)	<0.001
	…the whole thing is only about profit for the pharmaceutical industry		321 (22)	223 (16.1)	<0.001
COVID-19 vaccination is harmful because…	...long-term risks are not known		665 (47)	635 (45.7)	n.s.
	…new vaccines pose additional risks to the RNA		306 (21)	329 (24)	n.s.
	...there are doctors who advise against it		308 (22)	331 (24)	n.s.
	…a compulsory corona vaccination with prioritization of certain groups will lead to major socio-political discussions		512 (36)	478 (34)	n.s.
			Total 179 (100)	Total 174 (100)	
Mandatory vaccination (of children) is unnecessary because….	…it is not effective		15.2%	14.9%	n.s.
	…the natural immune system is enough		17.4%	16.1%	n.s.
	…these diseases no longer exist		9.5%	10.4%	n.s.
	…the whole thing is only about profit for the pharmaceutical industry		19.6%	22.4%	n.s.
Mandatory vaccination (of children) is harmful because….	…the risk is greater than the protection		13.5%	13.2%	n.s.
	……the vaccines are not controlled enough		17.4%	20.1%	n.s.
	…there are doctors who advise against it		15.7%	17.9%	n.s.
	…there have been negative vaccine experiences in my family		11.2	21.3	0.011
What do you actually (in the light of the COVID-19 discussion) think about mandatory vaccination?	It is important that my children get the necessary protection		81.5%	68.4%	0.005
	It is important to guarantee herd immunity		73.6%	59.5%	0.005
	I’m worried about the decline in the obligatory vaccination due to the pandemic		44.1%	32.4%	0.023
I would vaccinate my child because…	…I want to protect my child		73.7%	73.6%	n.s.
	…it is my parental responsibility		59.2%	59.2%	n.s.

^1^ The *p*-values refer to chi-square tests for ordinal and nominal data. Abbreviation: n.s., not significant.

**Table 3 vaccines-12-00176-t003:** Predictors of agreement with the national vaccination plan in South Tyrol, Italy, in March 2021 and 2023 in a multivariate logistic regression analysis.

		Model 2021 N = 1425 Nagelkerkes R^2^ = 0.332			Model 2023 N = 1388 Nagelkerkes R^2^ = 0.315		
		Regression Coefficient B	*p*-Value	OR [95% CI]	Regression Coefficient B	*p*-Value	OR [95% CI]
Constant term		−1.657	0.007		−1.940	<0.001	0.144
Age			n.s.			n.s.	
Urban residency			n.s.				
Educational level			n.s.				
Not vaccinated for COVID-19 *		−2.245	<0.001	0.106 [0.072; 0.157]			
CAM consultation			n.s.		−0.351	0.027	0.704 [0.516; 0.962]
Altruism	Sum score		n.s.			n.s.	
Trust	Sum score trust in institutions	0.066	<0.001	1.068 [1.049; 1.087]	0.108	<0.001	1.115 [1.095; 1.134]
Native language	German ^†^		n.s.			0.002	
	Italian		n.s.	1.454 [1.061; 1.993]	0.524	0.003	1.689 [1.194; 2.390]
	Ladin		n.s.			n.s.	
	Other/more than one		n.s.			n.s.	
Economic situation ^#^			n.s.			n.s.	

The *p*-values indicate significant contributions of independent variables to the model. ^#^ categorical variable (better, equal, worse, do not know): equal is used as an indicator. ^†^ categorical variable used as an indicator; * 2021: I would vaccinate for COVID-19; 2023: vaccinated for COVID-19 at least 2 times. Abbreviation: n.s., not significant.

## Data Availability

The data presented in this study are available upon request from the corresponding author. The data are not publicly available for political reasons due to conspiracies and the ethnolinguistic nature of the information.

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
