# Peer review of "Vaccine Hesitancy and Public Mistrust during Pandemic Decline: Findings from 2021 and 2023 Cross-Sectional Surveys in Northern Italy"

_vaccines, 2024, doi:10.3390/vaccines12020176_

Round 1
Reviewer 1 Report
Comments and Suggestions for Authors
The article “Vaccine hesitancy and Public Mistrust During Pandemic Decline: Findings from 2021 and 2023 Cross-sectional surveys in Northern Italy” has been reviewed
Find below comments and suggested changes:
Abstract: This study examines vaccine agreements in South Tyrol, Italy, within distinct socio-cultural and linguistic contexts. Using data from the 2021 and 2023 ‘COVID-19 Snapshot Monitoring’ extended surveys, we assessed changes in attitudes towards COVID-19 and other vaccinesations during the second andfrom 2021 to 2023 when final years of the pandemic activity ceased. Multivariate logistic regression analysis was used to examine factors such as trust in institutions, language groups, and use of complementary and alternative medicine. The representativeness of the study is supported by good participation rates, ensuring a comprehensive view of attitudes towards vaccination in the region.
The results show a marked deterioration in vaccine acceptance shifting significantly by 2023, with an initial hesitancy rate of approximately 15% in 2021,shifting to ????? by 2023. A significant decrease in Lack of trust in health authorities and negative correlation with complementary and alternative medicine consultations were observed. The results highlight the complex nature of vaccine hesitancy in diverse regions such as South Tyrol, and underline the need for targeted communication strategies and trust-building initiatives to effectively reduce hesitancy. This study provides critical insights for the formulation of public health strategies in diverse sociocultural settings.
· A significant decrease? Was there a previous value published? If not change to: Lack of …..
Introduction:
Suggested changes
Line 33: COVID-19 vaccine have been rapidly developed
Line 55: The main objectives of the first and second surveys carried out in 2021 and 2023 in South Tyrol,
Line 67: Therefore a new survey was conducted in 2023 during …..
Line 68: The 2023 survey aimed to assess
Materials and Methods
Line 84 : Better give the exact number of invitations sent
Line 89 : Participants were invited by postal letter or electronic mail?
Results
· Figure 1 include labels at the axis of the graphic
· Heading on Table 2 Response (rather) agree? Why the ambiguity?
· Figure 3 change Labels to :
At work, on free time, with family, with friends, No problem
Discussion:
Line 287-288 : while general vaccination disagreement has significantly increased by 2023,
Refrences:Revise completeness for ref
Steininger, R. South Tyrol: A Minority Conflict of the Twentieth Century (Studies in Austrian and Central European History and Culture); New Brunswick, NJ: Transaction, 2009;
Landesinstitut für Statistik der Autonomen Provinz Bozen - Südtirol Covid-19: Einstellungen und Verhalten der Bürger. Jänner 2021; astatinfo; Landesverwaltung der Autonomen Provinz Bozen - Südtirol: Bolzano (BZ), Italia, 2021;

Comments on the Quality of English LanguageMinor changes for comprehensive reading
Reviewer 2 Report
Comments and Suggestions for Authors
1.This is a meaningful paper that can provide critical insights for the formulation of public health strategies in diverse sociocultural settings.
2.What is the basis for the independent variables introduced in Table 3? Table 1 shows that there is no significant difference in variables such as age, Chronic disease (s), Native Language, and CAM consultation between non heterosis and heterosis individuals. Where do the variables CAM consultation, Altruism, and Trust come from? Why are variables such as age and Chronic disease (s), Work in the health care sector, and Economic situation that are not statistically significant between 2021 and 2023 listed in Table 3? If it is a covariate, specific statistical analysis results should also be listed.
3.From Figure 1, it can be seen that the vaccination rate of the research subjects in 2023 is already relatively high. Is this also a factor that causes vaccine hesitancy? Can the vaccination situation be included in the analysis of influencing factors in Table 3?
4.In 2023, the COVID-19 Pandemic Decline may cause most people to stop caring about vaccination rather than public Mistrust, which may be the cause of Vaccine Hesitancy that needs to be analyzed in the discussion.
5.The expression should be consistent before and after. (Table 1: Native Language, Table 3: Mother tongue).
6.The table should be in a three line table format.
7.Moderate editing of English language required.
Comments on the Quality of English Language
Moderate editing of English language required.
Round 2
Reviewer 2 Report
Comments and Suggestions for Authors
The author has answered all questions and made corresponding modifications and responses to them, and agrees to accept in present form.
Author Response
Thank you! Kind regards